# A Hydrodynamic Approach to the Study of HIV Virus-Like Particle (VLP) Tangential Flow Filtration

**DOI:** 10.3390/membranes12121248

**Published:** 2022-12-09

**Authors:** Tobias Wolf, Jamila Rosengarten, Ina Härtel, Jörn Stitz, Stéphan Barbe

**Affiliations:** 1Research Group Process Engineering, Faculty of Applied Natural Sciences, TH Köln—University of Applied Sciences, Campusplatz 1, 51379 Leverkusen, Germany; 2Institute of Technical Chemistry, Leibniz University Hannover, Callinstraße 5, 30167 Hannover, Germany; 3Research Group Pharmaceutical Biotechnology, Faculty of Applied Natural Sciences, TH Köln—University of Applied Sciences, Campusplatz 1, 51379 Leverkusen, Germany

**Keywords:** virus-like particle, tangential flow filtration, particle polarization layer, downstream process

## Abstract

Emerging as a promising pathway to HIV vaccines, Virus-Like Particles (VLPs) have drawn considerable attention in recent years. A challenge of working with HIV VLPs in biopharmaceutical processes is their low rigidity, and factors such as shear stress, osmotic pressure and pH variation have to be reduced during their production. In this context, the purification and concentration of VLPs are often achieved by means of Tangential Flow Filtration (TFF) involving ultrafiltration hollow fiber modules. Despite the urgent need for robust upscaling strategies and further process cost reduction, very little attention has been dedicated to the identification of the mechanisms limiting the performance of HIV VLP TFF processes. In this work, for the first time, a hydrodynamic approach based on particle friction was successfully developed as a methodology for both the optimization and the upscaling of HIV VLP TFF. Friction forces acting on near-membrane HIV VLPs are estimated, and the plausibility of the derived static coefficients of friction is discussed. The particle friction-based model seems to be very suitable for the fitting of experimental data related to HIV VLP TFF as well as for upscaling projections. According to our predictions, there is still considerable room for improvement of HIV VLP TFF, and operating this process at slightly higher flow velocities may dramatically enhance the efficiency of VLP purification and concentration. This work offers substantial guidance to membrane scientists during the design of upscaling strategies for HIV VLP TFF.

## 1. Introduction

The Human Immunodeficiency Virus (HIV) is the causative agent of Acquired Immunodeficiency Syndrome (AIDS), which if untreated results in the death of the patients, mostly caused by opportunistic infections [1]. Despite decades of research, no efficient vaccine against HIV has been developed yet. Virus-Like Particles (VLPs) play a prominent role in HIV vaccine development as they very effectively stimulate the immune system [2,3,4,5]. Due to the absence of genetic material, VLPs also exhibit a high safety profile.

The morphology of HIV VLPs is characterized by two components: a dense protein core formed by structural self-assembly of individual units from the HIV group-specific antigen (Gag) and a surrounding lipid bilayer originating from the host cell. HIV VLPs can be produced in the presence or absence of co-expression of the viral protease. The protease mediates the processing of the Gag precursor proteins by cleavage, facilitating the location of matrix proteins on the inside of the particle membrane and the formation of the typical cone-shaped core consisting of capsid proteins. In contrast, and in the absence of the viral protease, HIV VLPs remain immature as the Gag precursor proteins are not cleaved into their subunits [2,6,7]. Immature VLPs reveal a size of approximately 120 nm in diameter [8,9,10]. VLPs can be decorated with the envelope glycoproteins (Env) displayed on the surface, significantly improving the immunogenicity of HIV VLP vaccines [3,4,11]. Figure 1 schematically illustrates an immature HIV VLP either non-decorated (bald) or decorated with Env proteins.

While the expression cell line can affect the size of HIV VLPs, the size of a VLP population produced in one system can be considerably homogenous [12]. This fosters filtration as a preferred methodology to concentrate particles. A challenge of working with HIV VLPs in biopharmaceutical processes is their low rigidity. Factors such as shear stress, osmotic pressure and pH variation have to be reduced during processing [7]. In contrast, non-enveloped VLPs, e.g., derived from the Human Papillomavirus (HPV), are far less sensitive to these factors, which facilitates downstream processing. Besides the medical relevance of HIV VLPs, these particles are a preferred model for downstream studies as a large and growing collection of materials and methodologies is available for particle characterization and quantification [13,14,15,16]. From a macromolecular point of view, HIV VLPs may be regarded as highly functional and sensitive colloidal bionanoparticles.

Recently, Hengelbrock et al. [17] presented a process enabling the effective production, concentration and purification of HIV VLPs. It consists of the following six steps:Fed-batch cultivation of HIV VLP producing HEK293F cells (human embryonic kidney 293 cells),Removal of cells and large cell debris by depth filtration,Initial purification by tangential flow filtration,Further VLP purification by anion exchange chromatography,Concentration by tangential flow filtration,Formulation of HIV VLPs by lyophilization.

Tangential flow filtration (TFF) plays a major role in this process and usually involves ultrafiltration hollow fiber modules rather than flat sheet membrane cassettes. Despite considerable process optimizations, the production of nanoparticle-based biopharmaceuticals is still associated with high production costs and upscaling issues [18,19]. Both the high value and the above-mentioned sensitivity of HIV VLPs have a substantial impact on the design of dedicated TFF processes, which are typically carried out in void volume optimized modules involving ultrafiltration hollow fibers with low inner diameters (<1 mm). For this particular application, TFF is operated under laminar flow conditions by maintaining the shear rate below 4500 s^−1^ [20]. Furthermore, the particle concentration of the processed HIV VLP suspensions is very low with a mean value of $8.5\;\times \;10^{7}$ VLP/mL; this is equivalent to 0.017 vol% based on a spherical particle. These are quite unfamiliar operating conditions for the TFF processing of colloidal suspensions, in which mass transfer is often limited by the formation of a particle polarization layer (PP layer). It is therefore common practice to limit the development of PP layers by applying a high flow velocity and high shear rate resulting in turbulent flow conditions [21,22,23].

Despite the urgent need for robust upscaling strategies and further process cost reduction, very little attention has been dedicated to the identification of the mechanisms limiting the performance of HIV VLP TFF processes. It is well known that, during the TFF treatment of colloidal particles greater than 0.1 μm, the formation and further development of PP layers is mainly governed by hydrodynamic forces that result from near-membrane flow dynamics [24,25]. In this regard, the present contribution aims at approaching the HIV VLP TFF from a hydrodynamic point of view. The friction based model developed by Schock [24] is applied to estimate detachment, normal and friction forces acting on near-membrane HIV VLPs and the plausibility of the derived coefficients of friction is discussed. Furthermore, a methodology based on static coefficients of friction was developed as an optimization tool for HIV VLP TFF. The HIV VLP production methodology developed by Rosengarten et al. [10] and applied by Hengelbrock et al. [17] was chosen for this study for the sake of comparability. Systematic TFF single pass trials were performed with membranes exhibiting different Molecular Weight Cut-Off (MWCO) under different operating conditions (transmembrane pressure (TMP) and flow velocity).

## 2. Particle Friction-Based Model (PFBM) Developed by Schock [24]

Schock’s [24] approach relies on the separate and independent treatment of the axial and radial components of the complex near-membrane flow field by considering a horizontal laminar flow in axial direction due to the feed flow through the hollow fiber and a vertical laminar flow in the radial direction due to the permeate flow. It is a simplified representation of the complex physics at this location, which typically involves three-dimensional flow around nanoparticles and stagnation close to the membrane due to its flow resistance. Consequently, this approach can only lead to an approximation of the real near-membrane flow dynamics, and this uncertainty has to be considered when discussing the order of magnitude of derived characteristics such as forces and static coefficients of friction. This model is valid for flow conditions leading to the formation of a PP layer. Under these conditions and according to [24], a particle located at the surface of the membrane is immobile, being subject to force equilibrium as described in Figure 2. Consequently, in the axial direction *x*, the force balance at the center point of this particle can be formulated as follows.
(1)$$\left|{\vec{F}}_{D,a}\right|\;=\;|{\vec{F}}_{F}|$$

${\vec{F}}_{D,a}$ is the drag force resulting from the resistance of the particle in the axial fluid flow, and its magnitude is given by Equation (Equation 2).
(2)$$\left|{\vec{F}}_{D,a}\right|\;=\;c_{W,a}\;\cdot \;\frac{\rho }{2}\;\cdot \;u_{c,\mathrm{VLP}}^{2}\;\cdot \;\frac{\pi \;\cdot \;d_{P}^{2}}{4}$$

Considering this situation as a laminar flow around a sphere (Stokes flow), the drag coefficient $c_{W,a}$ can be calculated as follows:(3)$$c_{W,a}\;=\;\frac{24}{Re_{P,a}}\;=\;\frac{24\;\cdot \;\nu }{u_{c,\mathrm{VLP}}\;\cdot \;d_{P}}$$

Additionally, static friction ${\vec{F}}_{F}$, a force resisting the motion of the particle on the membrane surface, also acts on the particle. According to Equation (Equation 4), the magnitude of ${\vec{F}}_{F}$ depends on the magnitude of the radial drag force ${\vec{F}}_{D,r}$, which results from the fluid–particle interaction in the radial direction *y* and is caused by the permeate flux.
(4)$$|{\vec{F}}_{F}|\;=\;\mu \;\cdot \;\left|{\vec{F}}_{D,r}\right|$$

μ, the static friction coefficient, is mainly influenced by the roughness of the membrane and the contact area between both surfaces. The magnitude of ${\vec{F}}_{D,r}$ depends on the permeate flow velocity $u_{P}$ and is given by Equation (Equation 5).
(5)$$\left|{\vec{F}}_{D,r}\right|\;=\;c_{W,r}\;\cdot \;\frac{\rho }{2}\;\cdot \;u_{P}^{2}\;\cdot \;\frac{\pi \;\cdot \;d_{P}^{2}}{4}$$

$c_{W,r}$ can be calculated by considering a Stokes flow in radial direction.
(6)$$c_{W,r}\;=\;\frac{24}{Re_{P,r}}\;=\;\frac{24\nu }{u_{P}\;\cdot \;d_{P}}$$

Equation (Equation 1) can now be rewritten as follows.
(7)$$u_{c,\mathrm{VLP}}\;=\;\mu \;\cdot \;u_{P}$$

The axial component of the flow velocity at the center point of the particle in the laminar shear field was derived by Schock [24] (Equation (Equation 8)).
(8)$$u_{c,\mathrm{VLP}}\;=\;0.02\;\cdot \;Re^{1.75}\;\cdot \;\frac{\nu }{d_{H}}\;\cdot \;\left(\frac{d_{P}}{d_{H}}\right)$$

Finally, the permeate flux can be predicted by Equation (Equation 9).
(9)$$J\;=\;u_{P}\;=\;0.02\;\cdot \;\mu^{-1}\;\cdot \;Re^{1.75}\;\cdot \;\frac{\nu }{d_{H}}\;\cdot \;\left(\frac{d_{P}}{d_{H}}\right)$$

## 3. Materials and Methods

### 3.1. HIV Mos-1-Gag VLP Production and Clarification

All experiments were performed using the stable VLP producer cell line earlier described by Rosengarten et al. [10]. Cultivation was carried out in suspension with an initial start density of 3.5 × 10^5^ cells/mL to 5 × 10^5^ cells/mL in 300 mL Gibco™ Freestyle™ 293 expression medium in 1 L single-use PETG erlenmeyer flasks with a plain bottom (Thermo Scientific™ Nalgene™). The recombinant cell line was kept under a constant selection pressure of 15 μg/mL puromycin. Cultivation was performed at 37 °C, 8% CO_2_ and 135 min^−1^ (HERAcell 150i CO_2_ Incubator, Thermo Fisher Scientific, Waltham, MA, USA). VLPs were harvested from cell populations after cultivating for 3–4 days post inoculation. The cell suspension was clarified in a centrifugation step at 100× g for 5 min. Remaining cell debris was removed from the cell-free supernatant by passaging through a 0.45 μm PVDF filter (Carl Roth, Karlsruhe, Germany).

### 3.2. Tangential Flow Ultrafiltration

TFF trials were carried out as single pass trials with a KrosFlo^®^ KR2i TFF System (Repligen Corporation, Waltham, Massachusetts 02453, USA). The mass of permeate and the pressure of feed, retentate and permeate was recorded, as shown in Figure 3. A peristaltic pump is integrated in the KrosFlo^®^ system, which was calibrated gravimetrically with water as medium.

As the ultrafiltration (UF) membrane, a hollow fiber module (Repligen Corporation, Waltham, MA, USA) was used with an effective length of 20 cm, a fiber inner diameter of 0.5 mm and a fiber count of 6. This resulted in a surface area of 20 cm^2^, and the membrane material was modified polyethersulfone. On the basis of different studies [27,28], MWCOs of 100 kDa, 300 kDa, 500 kDa and 750 kDa of the membranes were chosen, where a high retention of the VLP was assumed. As hydrodynamic parameters, TMP and flow were varied as listed in Table 1. The TMP was limited due to the maximal operation condition of the pressure transducer. Additionally, the stability of the VLP was taken into account by maintaining shear rate values below 4500 s^−1^. This resulted in a volume flow range from 9 mL/min to 18 mL/min and led to a number of trials of 36 experiments—9 for each MWCO.

To eliminate the influence of a changing VLP concentration in the feed over the experiment, the trials were performed in a single pass. The particle concentration in the feedstock was around $8.5\;\times \;10^{7}$ VLP/mL or 0.017 vol%, determined via ELISA (Section 3.3.1). The volume of the feedstock was 250 mL depending on the flow and TMP, the filtration times varied between 10 min to 30 min, and a concentration factor 1.1 to 2.0 was reached.

At the beginning of every experiment series, the leak tightness was checked. Subsequently, the water permeability was determined. Therefore, the module was flushed with ultrapure water, and the permeate mass was recorded. To test different pressure levels, the valve position was fixed and the flow would increase from 10 mL/min to 40 mL/min in 10 mL/min steps. For every single volume flow, the flux was determined due to the change of the permeate mass. This procedure was performed before and after each trial.

After a pre-test of the system, around 250 mL of feedstock was used, and the module was flushed with VLP suspension. The suspension was circulated in a closed loop (permeate valve was closed). Tubing was then connected to the retentate bottle, the permeate valve was opened, and the pump was started.

At the end of filtration, the module was flushed with 0.5 mol L^−1^ NaOH. Before the next usage, the module was purged with ultrapure water. The module was stored in 0.5 mol L^−1^ NaOH. The complete removal of NaOH was checked with pH paper.

### 3.3. Analytics

#### 3.3.1. Enzyme-Linked Immunosorbent Assay (ELISA)

VLP samples were analyzed using a p24 ELISA kit (QuickTiter™ HIV Lentivirus Quantitation Kit (p24 ELISA); Biotrend, Cologne, Germany). The assay was performed as stated in the manufacturer’s instruction. The dilution of the samples was 1 to 10. The colorimetric readout was performed using the Multiskan FC (Thermo Fisher Scientific, Waltham, MA, USA) at a wavelength of 450 nm.

#### 3.3.2. Dynamic Light Scattering (DLS)

Particle size distribution was measured with a zetasizer nano ZS (Malvern Panalytical Ltd., Malvern, UK). Quartz glass cuvettes with an optical path length of 10 mm and a width of 4 mm were used. The sample volume was set to 350 μL. For particle size measurement, the number of runs was set to 15 with a duration of 10 s, and backscatter was detected at 173°. “Protein analysis” was chosen as the analysis method. The material properties for this method were provided by the zetasizer software and are listed in Table 2.

#### 3.3.3. Zeta Potential

For the measurement of the zeta potential, the zetasizer nano ZS (Malvern Panalytical Ltd., Malvern, UK) was used. The sample was placed in a DTS1070 capillary cell. The measurement was carried out at 25 °C, and the number of measurements was calculated automatically by the program. The material properties were the same as for DLS investigations (see Table 2), and as a model, the “Smoluchovski model” was chosen.

#### 3.3.4. Sodium Dodecyl Sulfate-Polyacrylamide Gel Electrophoresis (SDS-PAGE)

VLP samples were analyzed using sodium dodecyl sulfate–polyacrylamide gel electrophoresis (SDS-PAGE). In total, 50 μL sample was mixed with 16 μL Laemmli buffer (ROTI^®^Load 1, Carl Roth, Karlruhe, Germany) and incubated for 10 min at 96 °C. Then, 20 μL of sample and 10 μL molecular weight size marker (PageRuler™ 10 kDa to 180 kDa, Thermo Scientific™, Schwerte, Germany) were loaded on a precast polyacrylamid gel (12% Mini-PROTEAN^®^ TGX Stain-Free™ Protein Gels, Bio-Rad, Germany). As running buffer, a 1 to 10 dilution of ROTIPHORESE^®^10x (Carl Roth, Karlsruhe, Germany) and ultrapure water was used. The gel separation was started using a voltage of 60 V for 30 min. The separation was continued at 150 V for a further 60 min. Afterwards, the gel was washed in water for 30 min and stained using ROTI^®^Blue quick (Carl Roth, Karlsruhe, Germany) until protein bands became visible. For the gel image, the ChemiDoc XRS+ (Bio-Rad, Germany) was used.

#### 3.3.5. Transmission Electron Microscopy (TEM)

The concentrated and purified VLP samples were analyzed using a 200 kV JEOL JEM-2100PLUS transmission electron microscope (JEOL, Germany) and a negative stain method. The method of this is described by Rosengarten et al. [10].

### 3.4. Data Treatment and Estimation of Static Friction Coefficients

In order to reduce the effect of measurement fluctuations and ensure the extraction of steady state data, a data treatment strategy was designed. In this regard, only data were considered satisfying the selection criteria
(10)$$TMP\left\{\begin{array}{cc} <0.05 & \mathrm{for}\;\mathrm{experiments}\;\mathrm{with}\;TMP<\;0.5\;\mathrm{bar}\\ <0.15 & \mathrm{for}\;\mathrm{experiments}\;\mathrm{with}\;TMP>\;0.5\;\mathrm{bar} \end{array}\right.$$
to remove the run-in phase and run-out phase of the trials. First, the slope of TMP versus time *t* was estimated for five data points $m_{5}$ from $t_{\mathrm{i - 2}}$ to $t_{\mathrm{i + 2}}$. In addition, another slope $m_{2}$ was calculated between $t_{\mathrm{i+1}}$ and $t_{i}$ depending on the permeate mass against time. Then, the following steady state conditions was applied:(11)$$\begin{array}{c} |m_{5}|\;\leq \;0.001, \end{array}$$(12)$$\begin{array}{c} 0<\;|m_{2}|\;\leq \;0.3. \end{array}$$

In addition, the step size between each data point was adapted. The mean value of three data points was calculated to extend the time between each point from 10 s to 30 s. This give us the opportunity to minimize the error due to inaccuracies of measurement when weighing the permeate. According to this data treatment strategy, the stationary flux was defined as the flux measured once a constant TMP was reached and was calculated as the mean value during the last 2 min to 2.5 min or as the last five data points of each experiment. Based on Schock [24], the measured permeate flux *J* was simply plotted versus
(13)$$0.02\;\cdot \;Re^{1.75}\;\cdot \;\frac{\nu }{d_{H}}\;\cdot \;\left(\frac{d_{P}}{d_{H}}\right),$$
and we then performed a linear regression and extracted the static coefficient of friction μ from the slope
(14)$$m\;=\;\mu^{-1}.$$

As the particle diameter $d_{P}$, the median particle size if 155 nm was used, obtained from DLS measurements. Additionally, the inner diameter of 0.5 mm ($d_{H}$) was used as stated by the manufacturer of the fibers. Due to the permeate flux, the volume flow over the fiber length changed. Therefore, the Re was calculated out of the mean volume flow between the inlet and outlet of the fiber. With Equation (Equation 14) and the fitted slope *m*, the static friction coefficient μ was calculated, and the friction force was determined with the static friction coefficient μ. For this purpose, Equations (Equation 5) and (Equation 6) were inserted into Equation (Equation 4)
(15)$$F_{F}\;=\;3\pi \;\cdot \;\eta \;\cdot \;J\;\cdot \;d_{P}\mu .$$

## 4. Results and Discussion

### 4.1. VLP Formation and Characterization

The formation of spherical intact HIV VLPs was confirmed by TEM (Figure 4a), and the corresponding particles exhibited a mean size of 155 nm (Figure 5) and a zeta potential of −28.6 mV. The colloidal VLP suspension used in this work had a concentration of $8.5\;\times \;10^{7}$ VLP/mL corresponding to a volume concentration of 0.017 vol%.

### 4.2. HIV VLP TFF Trials

Pure water permeability was determined prior to HIV VLP TFF trials for each membrane (Figure 6) and, as expected, water permeability linearly increases with TMP. Surprisingly, the 500 kDa membrane exhibited a lower water permeability than the 100 kDa and 300 kDa membranes. Each membrane was cleaned with 0.5 mol L^−1^ NaOH after completion of the corresponding HIV VLP TFF series. After rinsing, pure water permeability was measured again, and the initial water permeability behavior could be recovered with the exception of the 750 kDa membrane. The water permeability of this membrane decreased by approx. 20%, which could result from irreversible fouling. As shown in Figure 4b, precursor proteins p55-Gag (55 kDa), known as a marker for the presence of HIV VLPs (s. Section 1) [10], could not be detected in the permeates, indicating that high HIV VLP rejection was achieved during the TFF trials. Furthermore, no change in particle size distribution was observed during the DLS analysis of the concentrates generated during the HIV VLP TFF trials. This observation confirms the reported stability of HIV VLPs up to shear rate values of 4500 s^−1^ [17,27,29].

Depending on the experimental conditions, the concentration factors achieved in this study were in the order of 1.1 to 2.0, and it should be noted that the present investigation mainly focuses on the beginning of the TFF-based HIV VLP concentration. In comparison with pure water permeability values, a severe decrease of the permeate flux was measured during the TFF treatment of HIV VLPs (Figure 7 and Figure 8). Furthermore, a clear influence of the TMP and the membrane’s MWCO on the permeate flux was not observed. Despite the low flow velocities applied in this study to maintain shear rate values below 4500 s^−1^, the performance of HIV VLP TFF seems to be dependent on the flow velocity. In this regard, higher permeate fluxes were measured at higher flow velocity values. Overall, these findings indicate that a PP layer controls the performance of HIV VLP TFF. Under this assumption, the impact of the membrane resistance would become negligible, as observed during the performed TFF trials. The missing evidence for the implication of TMP in the limitation of the filtration performance as well as the described positive effect of the flow velocity suggest that hydrodynamic effects mainly govern the PP layer formation.

### 4.3. Hydrodynamic Investigation of HIV VLP TFF

Nguyen [30] proposed a numerical method based on the calculation of a critical permeate flux $J_{\mathrm{crit}}$ in order to determine whether the flow conditions lead to the formation a PP layer or not. In this context, if the actual permeate flux is higher than $J_{\mathrm{crit}}$, the formation of a PP layer by means of particle deposition is expected. $J_{\mathrm{crit}}$ can be calculated by applying the formula presented by Nguyen [30]:(16)$$J_{\mathrm{crit}}\;=\;0.0807\;\cdot \;\frac{d_{P}^{2}}{\lambda \left(d_{P},\;c_{\mathrm{solid}}\right)}\;\cdot \;\frac{\tau_{W}^{1.5}\;\cdot \;\rho^{0.5}}{\eta^{2}}.$$

Similar to the approach developed by Schock [24], this equation was derived from a force balance set at the center point of the particle. The correction function $\lambda \left(d_{P},\;c_{\mathrm{solid}}\right)$ describes the interparticle interactions, which depend on both particle size and concentration. λ can only take values between 0 and 1 [31]; it was set to 1 in this work because of the low particle concentration of 0.017 vol% present in the feed. As shown in Figure 9, all permeate fluxes measured in this work are much higher than the corresponding critical fluxes. Consequently, it seems reasonable to assume that the above-mentioned TFF trials were performed under laminar flow conditions leading to the formation of a PP layer (Re < 127). This assumption legitimates the use of the particle friction-based model (PFBM) for fitting the experimental data collected in the framework of the present study.

The PFBM is based on static friction, which, in this particular case, would depend on the membrane surface roughness and the contact area between particle and membrane. For this reason, experimental data were sorted according to two different data classification schemes:*Four Class Scheme (4CS)*: Each membrane type (MWCO) corresponds to one class. Four classes in total (100 kDa, 300 kDa, 500 kDa and 750 kDa). Each class contains nine measurements.*One Class Scheme (1CS)*: All 36 measurements are enclosed in one single class.

Subsequently, one μ was determined for each class (considering 4CS & 1CS) by applying the methodology presented in Section 3.4. For the sake of validation, the class-dependent μs were used to predict the permeate flux of each individual measurement. A comparison between predicted and measured permeate fluxes is shown in Figure 10. Overall, the model predictions are in good agreement with the collected experimental data, and most predictions lie within a ±30% deviation when compared to the experimental data. Consequently, the PFBM seems to be appropriate for fitting experimental data collected from HIV VLPs TFF trials. The 4CS led to an improvement of the fitting accuracy, but it is not clear whether differences in membrane surface roughness can explain this observation. An increased fitting accuracy may simply result from a higher number of classes.

The accurate measurement of friction coefficients is the object of tribology and requires, in the case of nanoparticles, highly sophisticated methods usually involving Atomic Force Microscopy (AFM) and the application of normal forces in the order of 0.1 nN to 100 nN on individual nanoparticles [32,33,34,35,36,37,38]. The accurate measurement of μs during HIV VLP TFF is not the objective of this work. Near-wall flow calculations are always affected by a substantial uncertainty, and accurate μ values can hardly be determined by means of hydrodynamic calculations. Furthermore, the PFBM relies on a simplified representation of the near-membrane flow dynamics (s. Section 2), and derivate parameters can only be approximated. The aim of this research is the development of a practical approach for the optimization and the upscaling of HIV VLP TFF. The primary benefit of the apparent μs determined with the help of the PFBM is not of a tribological nature. These coefficients should be rather regarded as key parameters that allow for process optimizations and upscaling projections.

It is nevertheless necessary to compare the order of magnitude of both the μs and the friction forces estimated by PFBM fittings with tribological data in order to evaluate the suitability of an approach based on the near-membrane static friction of HIV VLPs in a flow field. In the literature, any μ values for HIV or HIV VLPs are reported, but a good overview of μs for common materials and materials combinations can be found in [39]. Depending on surface conditions (smooth or rough, clean or dirty, lubricated or dry), values vary from 0.02 (snow and PTFE) to 1.4 (aluminum and aluminum). Figure 11 shows a comparison of the class-dependent μs determined in this study. When applying the 4CS, the PFBM predict μs between 4.8 and 11.6, while a μ of 7.8 is predicted with the 1CS. These values are above the upper bound of μs reported in the literature, and the apparent μs derived from PFBM fittings seem to be overestimated, but the corresponding values are not far away from the order of magnitude of the reported μs. Furthermore, friction forces ranging from 0.25 nN to 1.11 nN were derived from the determined μs and are very well comparable with friction forces measured during the tribological investigation of nanoparticles [32,33,34,35,36,37,38]. These findings do not prove that near-membrane friction effects limit the performance of HIV VLP TFF, but they are very suggestive and should encourage the further development and practice of the presented approach.

From a technical point of view, an optimization strategy can be drawn from the friction-based approach by using the apparent μs for prediction purposes. Following this idea, permeate fluxes were predicted for different experimental conditions (different flow velocities and inner diameters of hollow fiber membrane) by maintaining shear rate values below 4500 s^−1^. The PFBM was fed with the calculated μs in order to estimate the boundaries of the optimization potential for the HIV VLP TFF presented in this work. As shown in Figure 12, the PFBM predicts considerable room for improvement of this filtration. An appreciable permeate flux increase could be achieved at slightly higher flow velocities as well as by increasing the inner diameter of the hollow fiber membrane.

## 5. Conclusions

This work offers, for the first time, an approach to the optimization and upscaling of the HIV VLP TFF. Based on the physical properties of VLP dispersions, we identified the PFBM developed by Schock [24] as a theoretical framework or a possible perspective to examine the formation of a PP layer during HIV VLP TFF. This theoretical framework requires assumptions regarding near-membrane flow dynamics, which are not in full agreement with the real phenomena occurring at this location. Consequently, due to this these assumptions and measurement error, the derived coefficients of friction can only be approximated and are therefore called **apparent coefficients of friction**.

As mentioned above, our work should not be regarded as a tribological study leading to the accurate measurement of friction coefficients but rather as an approach to generate key process parameters (apparent coefficients of friction), which in combination with the PFBM allows for fitting experimental data and making upscaling projections as shown in Figure 12. In this regard, the practical and technical benefit of the present contribution is substantial and meaningful. It will guide us during the design of optimization and upscaling strategies for HIV VLP TFF. HIV VLPs are by nature extremely soft particles, which, similar to viruses, may exhibit compaction (higher contact area) and be subject to adhesion mechanisms that may lead to higher apparent coefficients of friction. Tribological investigations reported in the literature are related to much harder particles, which are not fully comparable with HIV VLPs. Unfortunately, we cannot find any tribological investigation for HIV VLPs.

## Figures and Tables

**Figure 1 membranes-12-01248-f001:**
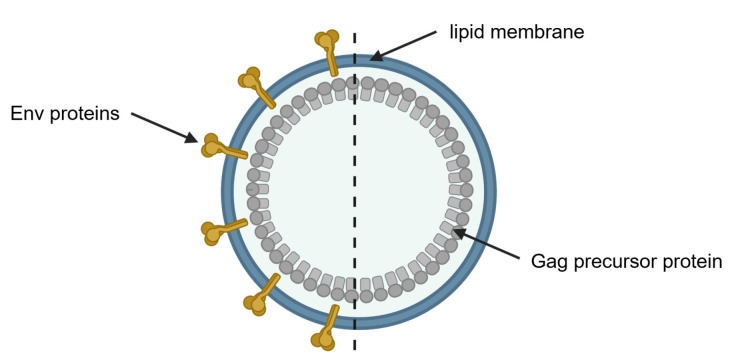
Schematic illustration of a membrane-enveloped HIV VLP. The protein core of immature HIV VLPs is composed of unprocessed Gag precursor proteins (grey) and surrounded by the lipid bilayer of the expression cell (blue). Envelope glycoproteins (Env) proteins are shown in yellow on the VLP, representing a decorated HIV VLP (left) and a bald VLP without decoration (right). Created with BioRender.com.

**Figure 2 membranes-12-01248-f002:**
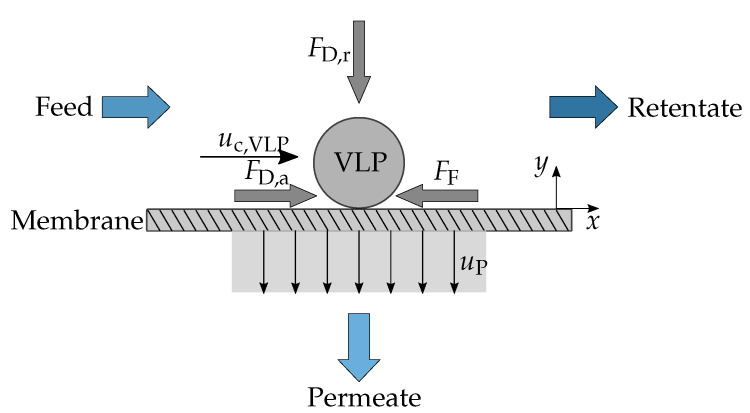
Schematic acting forces on a VLP located at the surface of the membrane according to Schock [24]. Figure adapted from [26].

**Figure 3 membranes-12-01248-f003:**
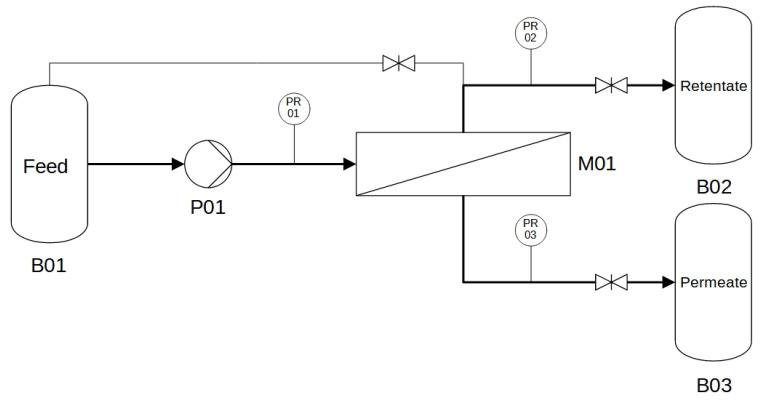
Process flow diagram of the experimental setup for UF experiments. B01 feed reservoir, B02 retentate reservoir, B03 permeate reservoir, P01 peristaltic pump, M01 hollow fiber module, PR pressure sensor.

**Figure 4 membranes-12-01248-f004:**
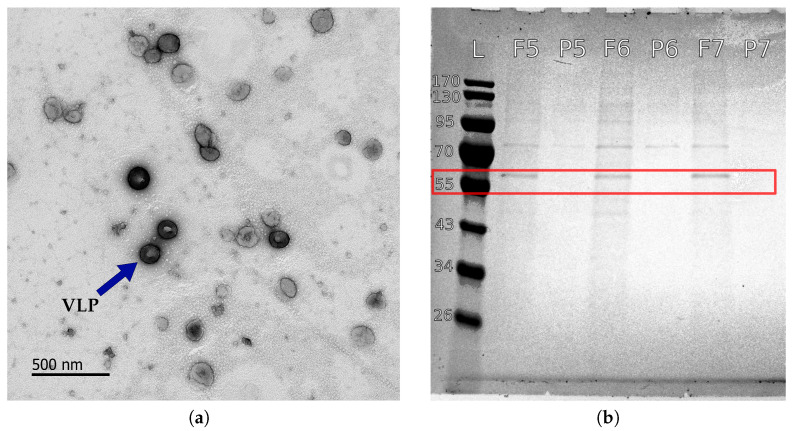
TEM image of a purified VLP suspension (**a**) and an SDS-Page of trail 5 to 7 feed (F) and permeate (P) sample after UF. The Mosaic p55-Gag precursor proteins in VLP are shown (red frame) (**b**). L represents a marker ladder.

**Figure 5 membranes-12-01248-f005:**
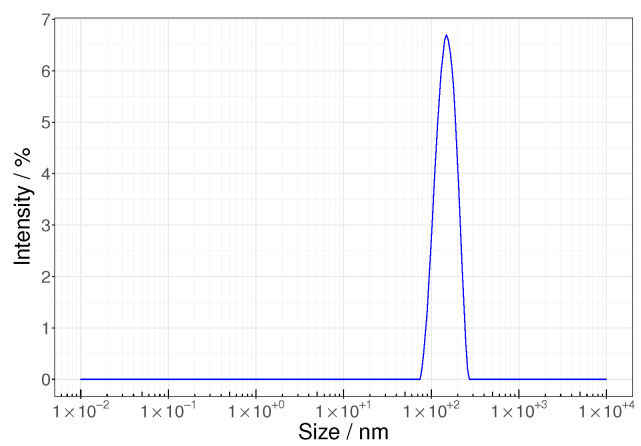
Particle size distribution of a VLP suspension after clarification.

**Figure 6 membranes-12-01248-f006:**
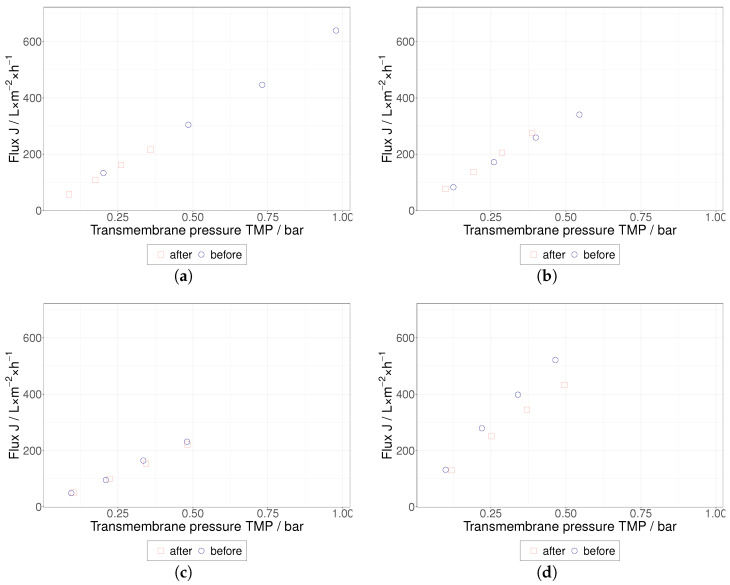
Flux of water over the TMP for each membrane was measured before and after tangential flow filtration trials; (**a**) 100 kDa, (**b**) for 300 kDa and (**c**) 500 kDa and (**d**) for 750 kDa MWCO.

**Figure 7 membranes-12-01248-f007:**
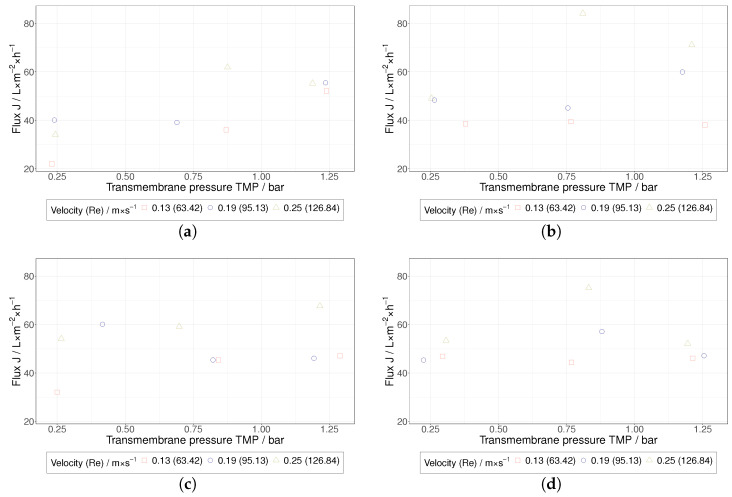
Stationary flux of the TFF treatment of HIV VLPs over the TMP for the different MWCOs; (**a**) 100 kDa, (**b**) 300 kDa, (**c**) 500 kDa, (**d**) 750 kDa.

**Figure 8 membranes-12-01248-f008:**
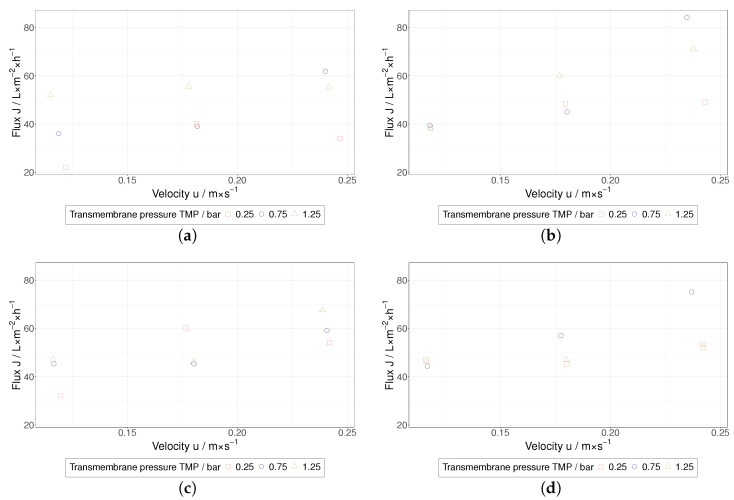
Stationary flux of the TFF treatment of HIV VLPs over the velocity for the different MWCOs; (**a**) 100 kDa, (**b**) 300 kDa, (**c**) 500 kDa, (**d**) 750 kDa.

**Figure 9 membranes-12-01248-f009:**
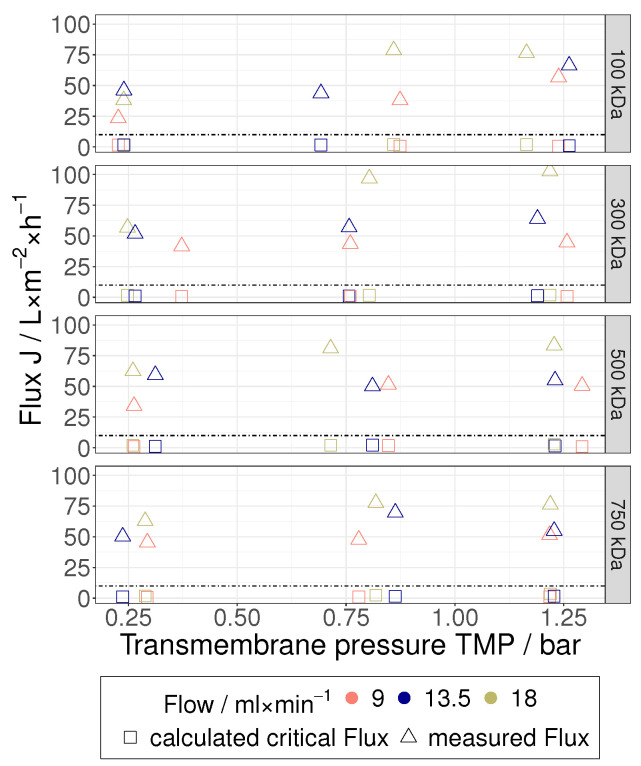
Calculated critical flux $J_{\mathrm{crit}}$ depending on the TMP split for the different MWCOs in comparison to the determined flux.

**Figure 10 membranes-12-01248-f010:**
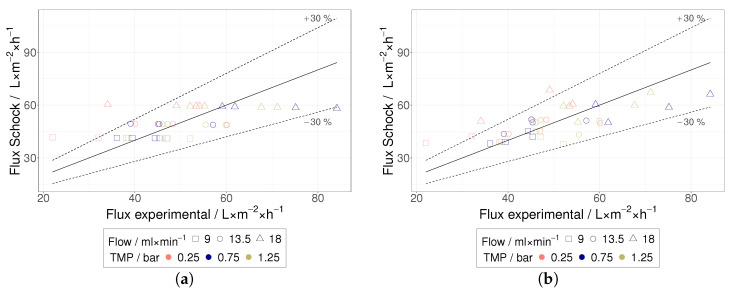
Experimental flux versus calculated flux according to Schock [24]; as function over u (**a**) for entire series of tests (1CS) and (**b**) for each MWCO (4CS). The straight line means a deviation of experimental versus model values by 0%, while the dotted lines show a deviation of ±30%.

**Figure 11 membranes-12-01248-f011:**
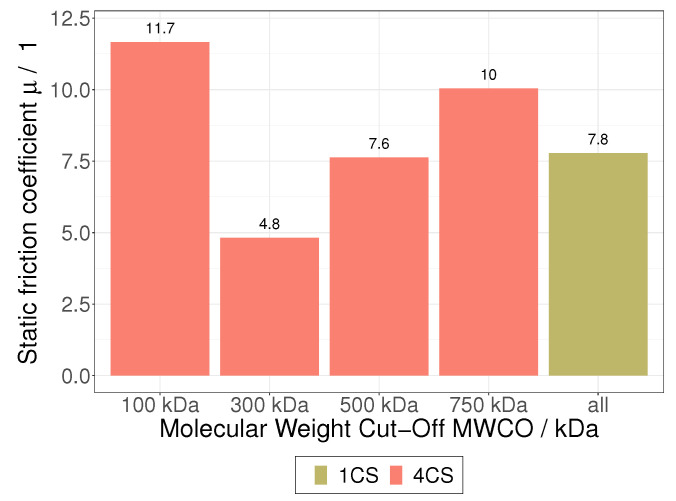
Comparison static friction coefficient μ of Schock for 4CS and 1CS.

**Figure 12 membranes-12-01248-f012:**
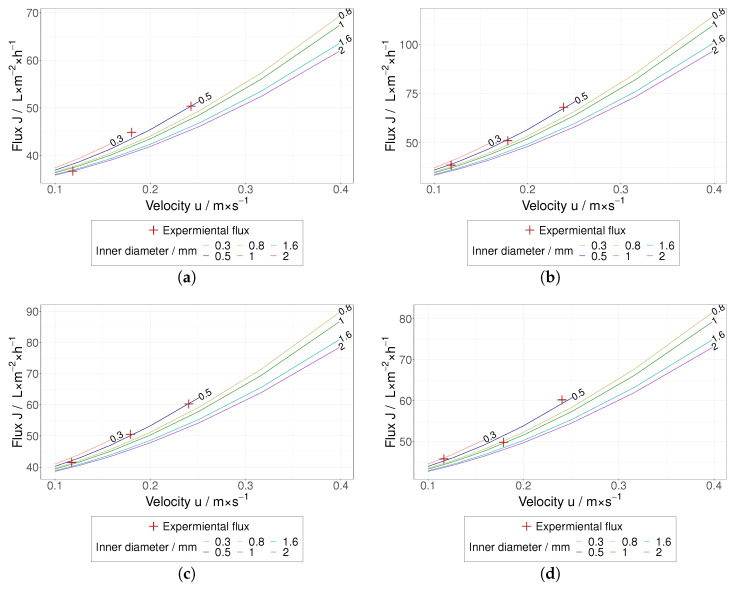
Flux over the velocity *u* for different $d_{H}$ from 0.3 mm to 2 mm based on Schock for the five different CS. The curves are limited for a $\gamma_{W}$ of 4500 s^−1^ and a Re of 2300; (**a**) 4CS 100 kDa, (**b**) 4CS 300 kDa, (**c**) 4CS 500 kDa, (**d**) 4CS 750 kDa, (**e**) 1CS.

**Table 1 membranes-12-01248-t001:** Hydrodynamic parameter of the UF.

Parameter	Levels		
TMP	0.25 bar	0.75 bar	1.25 bar
Volume flow	9 mL/min	13.5 mL/min	18 mL/min
Velocity *u*	0.13 ms^−1^	0.19 ms^−1^	0.25 ms^−1^
Reynolds number Re	63.42	95.13	126.84
Wall shear rate $\gamma_{W}$	2037 s^−1^	3056 s^−1^	4074 s^−1^

**Table 2 membranes-12-01248-t002:** Materials properties for the analysis model of zetasizer software at 25 °C.

Label	Substance	Refractive Index	Viscosity
Solvent	Water	1.330	0.8872 mPa s
Material	Protein	1.450	./.

## Data Availability

Not applicable.

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
