# Peer review of "A Hydrodynamic Approach to the Study of HIV Virus-Like Particle (VLP) Tangential Flow Filtration"

_membranes, 2022, doi:10.3390/membranes12121248_

Round 1

Reviewer 1 Report

The article by Wolf et al. discusses the purification and concentration of virus-like particles by means of tangential flow filtration. The article is well prepared and easy to read. However, suggest a minor revision and some changes.

1. ) Page 5, Line 128-130: Please give a few more details about the membranes such as membrane material and pore size.

2.) Page 8, line 213: Why was there reduced flux/no fouling visible for the membranes with the lower MWCO? Please explain this behavior.

3.) Figure 7 & 8.: Please specify the figure caption. E.g. "flux during the TFF treatment of HIV VLPs over TMP..."

4.) Figure 9: It is very hard to read the figures in general (e.g. very small symbols and small font), but in Figure 9 and 10 it is almost impossible to see the symbols and read what is written on the right side.

Author Response

Dear reviewer,

thank you very much for helpful comments. We dedicated careful attention and tried to address all your points. Please find below our answers to your comments:

Point 1: Membranes used in this work were made from modified polyethersulfone and we added the Molecular Weight Cutt Off values for each membrane. Unfortunately, pore size values were made available by the manufacturer.

Point 2: It seems that the employed cleaning procedure was working well for these membranes.

Point 3: We updated the caption as suggested.

Point 4: We increased the size of symbols and font.

Reviewer 2 Report

The manuscript investigated the Tangential Flow Filtration (TFF) filtration purification and concentration of HIV VLPs by ultrafiltration hollow fiber modules from a hydrodynamic perspective, which is meaningful. However, there should be a lot of work need to do and the current version doesn’t meet the quality of Membrane based on the present writing, experimental results and organizations which need major revises. The major deficiencies are as follows:

Major comments:

1.      The basic format of the manuscript was not uniform, for example, the first line of each paragraph needs to be indented.

2.      In the abstract, the main conclusions and research significance of the current findings should be indicated.

3.      Page 5, Line 136: “The particle concentration in the feedstock was around 8.5 × 107 VLP/mL respectively 0.017 vol %”, Please provide the particle concentration test method.

4.      Picture 3 was not found in the whole manuscript, please add it.

5.      Some experiments in the 3.3 section, used "zetasizer nano ZS" and "TEM" without mentioning the manufacturer and brand.

6.      In the 3.2 section, the experimental method only mentioned the operation method under different TMP, without mentioning the different filtration velocity, operation methods and filtration times. Please reorganize.

7.      Page 7, Line 191: Please provide an exact definition of " stationary flux", is it the flux after each membrane has been filtered for the same time?

8.      In section 4.1, the concentration of a stable colloidal suspension could not be deduced from Figures 4 and 5.

9.      The legend "after/ before" in Figure 6 should be explained in the caption, and multiple experiments are required to add error bars. Additionally, the conclusion "the 500 kDa membrane exhibited a lower water permeability than the 100 kDa and 300 kDa membranes" need to repeat the experiment and re-verify, which was obviously different from the typically reported results in the literature.

10.  Figure 7 and Figure 8 had a single form and were difficult to understand. Captions need to be added for the explanation, please redesign. Additionally, the experimental design should follow the principle of single variable to conduct comparative analysis.

11.  The legend of Figure 10(a) is incomplete, and the caption needs to explain the meaning of the straight line and +30%, -30%.

12.  Page 11, Line 229-231: “The absence of a clear impact of the TMP on the filtration performance as well as the described positive effect of the flow velocity suggest that hydrodynamic effects mainly govern the PP layer”, the explanations were too affirmative, since there is no direct information to support this hypothesis.

13.  Page 12, Line 240-241 “it was set to 1 in this work because of the low particle concentration of 0.017 vol% present in the feed”, Please provide relevant literature support.

14.  Page13, Line 276-279: The results showed that the µs were between 4.8-11.7, which were much higher than the 0.02-1.4 reported in the relevant literature. It is questioned whether there was an error in the calculation method of the model, and the results were not convincing. In addition, the conclusion section explained “This overestimation can be explained by the imprecision of near-wall flow calculations, the simplified flow representation assumed during the development of the PFBM and measurement errors” This made the research meaningless for practical application and contradicted the research content.

Minor comments:

1.      In the whole manuscript, please avoid using “we”.

2.      Page 1, Line 5: “the purification and concentration of VLPs is often achieved by means…” please change “is” to “are”.

3.      Page 3, Line 70: “the formation and further development of a PP layers is…” please revise “a PP layers”.

4.      Page 5, Line 133: “leads to a number of trails of…” please change “trails” to “trials”.

5.      Page 5, Line 148: “Before the next usage the modul was purge with ultrapure water” please change “purge” to “purged”.

6.      Page 5, Line 149: “The module war stored in 0.5 mol L−1 NaOH”, please change “war” to “was”.

7.      Page 7, Line 192: contained the repeated word " the".

8.      Page 7, Line 188: a comma should be added after " In addition ".

9.      Page 12, Line 237: “Similarly to the approach developed by Schock” please change “Similarly to” to “Similar to”.

10.  Page 14, Line 258: “The 4CS lead to an improvement of the fitting accuracy…” please change “lead” to “led”.

11.  Page 17, Line 290: “hollow fibre membrane” please change “fibre” to “fiber”.

Author Response

Dear reviewer,

thank you very much for the careful reading of our manuscript and your helpful and constructive comments. We tried to address all your points. Please find below our answers to your comments:

Major comments:

Point 1: We updated the manuscript format as suggested

Point 2: Main conclusions and research significance are now indicated in the abstract.

Point 3: Particle concentration measured via ELISA. We updated the manuscript as suggested.

Point 4: Done

Point 5: We added details regarding the manufacturers of this equipment.

Point 6: We performed randomized trials by setting one flow rate corresponding to one flow velocity and one TMP. We reread section 3.2 and think that it the operation method is appropriate.

Point 7: The stationary flux was defined as the flux measured once constant TMP was reached and was calculated as the mean value during the last 2 to 2.5 min of the period. This sentence was added to section 3.4.

Point 8:  We did not intend to link this conclusion with Figures 4 and 5. We changed it in the following way: “The colloidal VLP suspension used in this work had a concentration … “

Point 9: We updated the caption of Figure 6, which now explains after/before. We only got access to the Molecular Weight Cut Off values (MWCO), which gives information about the rejection behaviour of an ultrafiltration membrane. Nevertheless, these values cannot be directly correlated with pore size values. It is true that ultrafiltration membranes with higher MWCOs tend to have high permeate flux. However, fluctuations in permeability between production batches are usual. For these membranes, the manufacturer did not give any information related to a minimal permeate flux in the corresponding production specification sheet.

Point 10: We added relevant details in the caption and increased the size of symbols and font in order to improve the clarity of Figures 7 & 8. We did not understand your comment related to the principle of variable. Could you please give us a more detailed description?

Point 11: Caption was updated with explanations related to the straight line and +30%, -30%.

Point 12: Yes you’re right, we turned that sentence in this way: The missing evidence for the implication of TMP in the limitation of the filtration performance as well as the described positive effect of the flow velocity suggest that hydrodynamic effects may govern PP layer formation during HIV VLP TFF.

Point 13: We provided relevant literature regarding this point (Tam et al.).

Point 14: This work offers for the first time an approach to the optimization and the upscaling of the tangential flow filtration of HIV VLP (HIV VLP TFF).  Based on the physical properties of VLP dispersions, we identified the particle friction based model (PFBM) developed by Schock as a theoretical framework or a possible perspective to examine the formation of a particle polarization layer (PP layer) during HIV VLP TFF. This theoretical framework requires assumptions regarding near-membrane flow dynamics, which are not in full agreement with the real phenomena occurring at this location. Consequently, due to this these assumptions and measurement error, the derived coefficients of friction can only be approximated and are therefore called apparent coefficients of friction.

This does not mean that the determined coefficients are meaningless. As mentioned in the manuscript (line 261-265), our work should not be regarded as a tribological study leading to the accurate measurement of friction coefficients but rather as an approach to generate key process parameters (apparent coefficients of friction), which in combination with the PFBM allows for fitting experimental data and make upscaling projections as shown in Figure 12. In this regard, the practical and technical benefit of the present contribution is substantial and meaningful. It will guide us during the design of optimization and upscaling strategies for HIV VLP TFF. HIV VLPs are by nature extremely soft particles, which similar to viruses may exhibit compaction (higher contact area) and be subject to adhesion mechanisms that may lead to higher apparent coefficients of friction. Tribological investigations reported in the literature are related to much harder particles, which are not fully comparable with HIV VLPS. Unfortunately, we cannot find any tribological investigation for HIV VLPS.

Minor comments: We followed your advices and corrected the manuscript according to your suggestions.